# Psychophysiological Parameters Predict the Performance of Naive Subjects in Sport Shooting Training

**DOI:** 10.3390/s23063160

**Published:** 2023-03-16

**Authors:** Artem Badarin, Vladimir Antipov, Vadim Grubov, Nikita Grigorev, Andrey Savosenkov, Anna Udoratina, Susanna Gordleeva, Semen Kurkin, Victor Kazantsev, Alexander Hramov

**Affiliations:** 1Baltic Center of Neurotechnology and Artificial Intelligence, Immanuel Kant Baltic Federal University, Kaliningrad 236041, Russia; 2Neuroscience and Cognitive Technology Laboratory, Innopolis University, Kazan 420500, Russia; 3Neurodynamics and Cognitive Technology Laboratory, Lobachevsky State University of Nizhny Novgorod, Nizhny Novgorod 603022, Russia

**Keywords:** EEG, training, brain, sport shooting, biomarkers

## Abstract

In this study, we investigated the neural and behavioral mechanisms associated with precision visual-motor control during the learning of sport shooting. We developed an experimental paradigm adapted for naïve individuals and a multisensory experimental paradigm. We showed that in the proposed experimental paradigms, subjects trained well and significantly increased their accuracy. We also identified several psycho-physiological parameters that were associated with shooting outcomes, including EEG biomarkers. In particular, we observed an increase in head-averaged delta and right temporal alpha EEG power before missing shots, as well as a negative correlation between theta-band energies in the frontal and central brain regions and shooting success. Our findings suggest that the multimodal analysis approach has the potential to be highly informative in studying the complex processes involved in visual-motor control learning and may be useful for optimizing training processes.

## 1. Introduction

Sport shooting represents a complex sensorimotor process requiring a high level of visuospatial work. Shooting sports demand athletes maintain a good psychological state [1], stress control ability, and the ability to efficiently allocate cognitive resources (e.g., attention) during the shooting and aiming period [2]. As a consequence, training in sport shooting is a non-trivial challenge that often requires an individualized approach, especially in a sport with such high achievements [3]. The identification of the psychological and psychophysiological profile of a successful shooter is associated with superior performance, and the building of a training strategy focused on achieving the quickest achievement of this state could help in solving this problem.

The development of modern, compact, and mobile devices for multimodal monitoring of human physiological parameters and the rapid progress in neuroimaging technologies makes it possible to monitor the current state of an athlete concurrently with their behavioral performance to form representations of a successful profile. Currently, research in this direction is mainly focused on identifying biomarkers of the cardiovascular and respiratory systems operation [2], gaze behavior [4], as well as EEG biomarkers of successful shooters [3]; this research is generally based on the comparison of novice shooters with professional athletes [4,5,6]. However, with this approach, it is impossible to obtain information about the “trajectory” of the transformation from a novice shooter to a professional. A promising experimental paradigm from this point of view is the paradigm aimed at comparing successful and unsuccessful attempts at sport shooting training sessions in a naïve group. This approach makes it possible to identify what distinguishes successful attempts in novice athletes at the level of physiological parameters and EEG characteristics and to investigate the effect of training in detail. Recently, a trend for research in this direction has emerged. Note the study [7] that revealed EEG and kinematic biomarkers of precision motor control and changes in the neurophysiological substrates in naïve participants that may underlie motor learning during simulated marksmanship in immersive virtual reality.

However, many issues still remain unexplored. In particular, it is unknown exactly how and which physiological parameters and EEG characteristics change during sports shooting training; for example, which parameters correlate with shooting success and can thus claim to be biomarkers that are components of a professional athlete’s profile. Moreover, most studies generally examine the dynamics of one or two physiological parameters during shooting training (e.g., a study [2] utilized synchronized monitoring of EEG and electrocardiogram (ECG) to understand the mechanism of dual activation of the brain and heart in pistol athletes during shooting performances). At the same time, a deeper understanding of the relationship between physiological and psychological processes and training success can only be achieved by simultaneously considering as many physiological parameters as possible. Biomarkers of successful sport shooting should be searched not at the level of operation of individual subsystems of the human body but at the level of their joint operation and interaction; therefore, it is necessary to use multimodal registration of physiological parameters to solve this problem [2].

The present study takes a step toward solving the problems formulated. Here, we analyze multimodal data of subjects (EEG, ECG, electrooculogram (EOG), respiration activity (R), and fatigue tests) naïve to sport shooting training and study correlations between the psychophysiological parameters and shooting performance of the subjects. The special aspect of this study is the analysis of changes in fatigue levels during training and its effect on shooting success.

From a fundamental point of view, sport provides an ideal model for understanding neural adaptations associated with intensive training over time. We believe that the increased knowledge of links between physiological parameters, brain activity, and behavior characteristics will help to improve the effect of sport shooting training and thus enhance sports performance.

## 2. Materials and Methods

### 2.1. Participants

Experimental study included 21 healthy volunteers (all male, age 19–25, with an average age of 21 and a standard deviation of ∼1.5, right-handed). All subjects had no diseases that affected sight or locomotor functions. A healthy lifestyle was advised for the subjects prior to the experiment, which included sufficient night rest, no alcohol or drug consumption, and moderate physical activity. All subjects were volunteers; they were informed about the details of the study prior to participation, were able to ask related questions, and after that, provided informed consent. All participants were naïive to sport shooting, so before the experiment, a trained coach explained to them the basic principles and safety regulations. This study was conducted according to the guidelines of the Declaration of Helsinki and approved by the Ethics Committee of Lobachevsky University (Protocol №3 from 8 April 2021).

### 2.2. Experimental Setup

During the experiment, we recorded multimodal data from a subject: EEG, EOG, ECG, respiration activity (R). The placement of all sensors is shown in Figure 1A. All these signals were recorded by a wearable EEG recorder “Encephalan-EEGR-19/26” (Medicom MTD, Russia). The sampling rate for all types of data was 250 Hz. For EEG recording, we used 31 Ag/AgCl electrodes placed on the scalp according to the international scheme “10-10” (Figure 1A, grey circles). Other biological signals, besides EEG, were acquired through additional POLY channels of “Encephalan”. To record EOG, we used 2 electrodes (“EOG+” and “EOG-”) above and below the right eye (Figure 1A, green circles). The resulting EOG signal was calculated as the difference between these two signals. The right eye was chosen as it is usually the one used while aiming the shot. To record ECG, we placed 1 electrode on the subject’s back near the left scapula (Figure 1A, blue circle). Respiration activity was collected via a belt-shaped sensor wrapped around the subject’s chest (Figure 1A, white stripe). The stretching and contraction of the belt are associated with the expansion and compression of the thorax during respiration.

When choosing the sensors’ placement, we tried not to restrict the subject’s movement and, at the same time, tried to minimize the influence of this movement on the recorded signals. The “Encephalan” device was placed on the small of the back with a special belt, and all wires from the device to the sensors were tightly packed together and fixed on the back. The “Encephalan” was connected to the PC through Bluetooth, so this connection provided no additional restriction on the subject’s movement.

### 2.3. Experimental Procedure

The shooting was performed from an upright position, as illustrated in Figure 1B. For the experiment, we chose an air rifle with characteristics close to the real rifle used by sportsmen in biathlons. The rifle’s dimensions are 1010/270/85 mm (length/height/width), and its weight is ∼4 kg. The rifle uses a 4.5 mm caliber with a 5-round magazine and open sights. Since this was an air rifle, the recoil was not significant. Protective gear included shooting glasses but not headphones. The subjects shot at 5 separate targets at a distance of 10 m. The targets mimicked the ones used in biathlons at a distance of 50 m, so the targets in the experiment were properly scaled in size. The subject had visual and audial feedback after each shot—the successfully struck target changed color and provided distinct sound.

The experimental session included 21 series of shootings with Multidimensional Fatigue Inventory (MFI-20) [8], and the NASA Task Load Index (NASA-TLX) [9] tests before the first and after the last series correspondingly (see Figure 1C). The first series was treated as a test, so these results were excluded from further analysis. Each series included the following steps (see Figure 1D):**Preparation**—the subject received the rifle loaded with 5 bullets from the assistant and assumed shooting stance;**Shooting**—the subject performed 5 shots at 5 targets in any order;**Completion**—the subject quit shooting stance and handed the rifle back to the assistant for reloading;**VAS**—the subject passed a visual analog scale (VAS) test [10] for fatigue estimation;**Rest**—the subject rested for 60 s before the next series.

To assess changes in some behavioral and physiological characteristics throughout the experiment, we turned 20 series of shootings into 4 blocks. This was done by averaging results of 5 consecutive series, i.e., 1–5, 6–10, 11–15, 16–20.

MFI-20 is the test aimed at assessing a subject’s fatigue through self-report. This test includes 20 questions covering 5 dimensions of fatigue: Physical, Mental, and General Fatigue, as well as Reduced Activity and Motivation. NASA-TLX is another instrument to measure fatigue, but in this case, task-induced fatigue. The test includes several scales and their paired comparisons that help to assess 6 factors: Physical, Mental, and Temporal Demand, as well as Effort, Frustration, and Performance. VAS is used to subjectively measure the fatigue of the subject in his current state. Self-report is performed with the help of a continuous scale, on which the subject chooses the value of his current fatigue. The scale varies between “the lowest” and “the highest fatigue”. For all fatigue-assessment tests, we used a tablet computer.

We considered several factors during statistical analysis:“block”—reflects the course of the experiment, includes blocks 1–4;“phase”—reflects the subject’s type of activity in the experiment, including rest and shooting;“result”—reflects successfulness on each shot, including hits and misses.

### 2.4. Data Processing

The goals of preprocessing procedure were the following: for EEG data—to obtain clear signals without noises and artifacts for further time-frequency analysis, for respiration, EOG, and ECG—to obtain signals clear enough for extracting desired features such as blink rate or heart rate.

For EEG preprocessing, we used Fieldtrip toolbox for MATLAB [11]. EEG signals were filtered with a band-pass filter (cut-off frequencies—1 and 70 Hz) and 50 Hz notch filter in preparation for further time-frequency analysis.

To remove eye- and heart-related activity artifacts from EEG, we used a method based on Independent Component Analysis (ICA). For this, we applied *ft_componentanalysis* with the method *runica*. We decomposed EEG data into a set of independent components, searched components with artifacts, removed them, and then restored EEG signals with the remaining components. To ensure data quality, we performed additional visual data analysis with *ft_rejectvisual*. We rejected trials of data and/or EEG channels with severe artifacts remaining after the ICA-based procedure. Most of these artifacts were related to the subject’s active movement. We removed “bad” trials from the dataset, while for “bad” channels, we performed a repairing procedure with *ft_channelrepair*.

We performed a time-frequency analysis of EEG signals using continuous wavelet transform (CWT) with Morlet mother wavelet function [12]. We considered wavelet power (WP) as $W_{n}\left(f,t\right)$, where n=1,2,…,N is the number of EEG channel (N=31 for the considered dataset), *f* and *t* are the frequency and time point. WP is one of the common CWT-based characteristics to describe the time-frequency structure of a signal [13].

To reduce the data dimensionality, we considered averaged CWT spectra. Firstly, we averaged WP over several areas in the cortex: frontal (F), central (C), parietal (P), occipital (O), left temporal (LT), and right temporal (RT) (see Figure 1E). Secondly, we averaged WP over commonly used frequency bands: delta (1–4 Hz), theta (4–8 Hz), alpha (8–13 Hz), and beta (13–30 Hz). In our research, we considered a 2-s time interval just before the subject pulled the trigger. So we additionally averaged WP over this time interval.

We used the NeuroKit2 software package to process signals obtained from the respiratory sensor. NeuroKit2 is an open-source Python package designed to process neurophysiological signals [14]. For primary processing and filtering of the incoming signal, we used a linear detrending method with subsequent application of a low-pass fifth-order IIR Butterworth filter at the frequency of 2 Hz. The procedure is based on the zero-crossing algorithm with the amplitude threshold described in [15]. Then, we determined peaks (beginning of exhalation) and valleys (beginning of inhalation) using different sets of parameters described in [15]. Next, we determined the breathing phase defined between “1” for inspiration (inhalation) and “0” for expiration (exhalation). Then, we calculated the instantaneous frequency of the signal (in “1/min”) from a series of peaks. It is calculated as “60/period”, where the period is the time between peaks. To interpolate the frequency over the entire duration of the signal, the monotone cubic interpolation method was used. We also calculated the average values of frequencies at different stages of the experiment. For this purpose, the instantaneous respiration rate was calculated for each session at the moments of shooting and rest; further, the obtained rate values were averaged and added up for each subject.

We analyzed EOG to detect eye movement and blinking using the methods of the software package MNE [16], which turned out to be the most effective for this problem. We used a default set of parameters for this method. Additionally, we obtained the values of the signal peaks, which correspond to the moments of the subject’s blinks. Next, we calculated the blink rate (in minutes) from the series of peaks as “60/period”. Monotone cubic interpolation method was used to interpolate the frequency for the entire duration of the signal. Then, the average values of blink rates at the moments of shooting and rest were obtained for each subject.

To process the ECG signal, we filtered the data using high-pass and low-pass filters in the 1–6 Hz range. Further, R-peaks, which are distinguished by high amplitude and frequency, were selected from the prepared signal. We calculated heart rate as the inverse of the R-R interval ($1/t_{R-R}$). All heart rate values for each individual step were averaged for each subject.

We have considered different time window scales for the analysis of heart rate, respiration rate, and blink rate. To find a difference between stages of the experiment (rest vs. shooting), we averaged heart rate, respiration rate, and blink rate in windows length equal to respective stages. The time length of windows for the resting stage is 60 s, but windows for the shooting stage have different lengths (average length of 22.5 s) because of different rates of shooting across the subjects and shooting stages. Additionally, we analyzed the influence of instantaneous (right at the moment of shot) RR on shooting results.

The main effects at the group level were evaluated via Repeated Measures Analysis of Variance (RM ANOVA). We considered “block”, “result”, “phase”, and cortical area as within-subject factors in those statistical tests where the influence of these factors was considered. The post hoc analysis used either paired samples *t*-test or Wilcoxon signed-rank test, depending on the samples’ normality. Normality was tested via the Shapiro–Wilk test. The group-level correlation analysis between all pairs of characteristic changes during the experiment, such as heart rate, respiration rate, characteristic of the brain activity, hit rate, and subjective fatigue, was performed using repeated measures correlation. Correlations between subjective tests (MFI-20, NASA-TLX) and shooting accuracy were searched using Spearman’s rank correlation coefficient. We used several open-source statistical packages in Python, such as Pingouin, SciPy, statsmodels, and a package called JASP for statistical analysis and results visualization.

## 3. Results

### 3.1. The Behavioral Data Analysis

The results of the assessing subject’s state before the experimental task with the MFI-20 test are shown in Figure 2A. The median values are low (less than 8 out of a possible 20) across all scales of MFI-20, which confirms that none of the subjects has asthenia of any type.

To assess the task-induced load, we used a NASA-TLX test, and the results are shown in Figure 2B. We found that the experimental task induces low temporal and mental loads, while the main load is caused by the effort to preserve a certain level of performance.

The results of the change in fatigue level during the task assessed with VAS after each series of shootings are shown in Figure 2C. We considered z-scored results of VAS for a more universal data presentation. We found a significant increase in fatigue from block to block, and post hoc analysis showed significant differences between all blocks of the experiment. However, absolute values for the induced increase in fatigue (i.e., the difference between fatigue at the beginning and at the end of the experiment) are close to 30 out of 100 (maximal value in the scale). We suggest that this result indicates a low overall increase in fatigue during the experiment.

We used the hit rate as a parameter for evaluating the success of performance. The subjects coped well with the task: ∼65% of the shots hit the target on average. We analyzed changes in hit rate over the course of the experiment and found a significant increase in hit rate (RM ANOVA: p<0.001). Post hoc analysis showed significant differences between the first and fourth blocks, as well as between the third and fourth blocks.

### 3.2. The Physiological Data Analysis

We analyzed changes in physiological characteristics during the experimental task, both in the resting and shooting phases.

#### 3.2.1. Heart Rate

We did not find significant changes in the heart rate during the experiment, as well as no significant differences between heart rates at rest and shooting phases. However, we found an interaction effect between factors “block” and “phase” (*p* = 0.000531). Post hoc analysis showed that there are significant differences in heart rate between blocks 1–3 and 1–4 in the rest phase (see Figure 3A). Additionally, we considered heart rate variability as another characteristic of heart activity but did not find significant changes.

#### 3.2.2. Respiration Rate

Then, we analyzed the dynamics of respiration rate and found the interaction effect between “block” and “phase” (*p* = 0.045) factors, while no changes were detected in respiration rate during the experiment and between the phases. In the post hoc analysis, we found a decrease in respiration rate during the shooting phase, but the statistical significance of these changes is near the accepted threshold (see Figure 3B). Further, we studied the effect of instantaneous respiration rate on shooting success and found a significant difference in the instantaneous respiration rate between misses and hits (*p* = 0.043, see Figure 3C).

#### 3.2.3. Blinking Rate

We have not found significant changes in the blinking rate during the experiment or any relationship between the blinking rate and the hit rate.

#### 3.2.4. Brain Electrical Activity

We analyzed changes in the electrical activity of the brain directly before each shot, both for the “block” and “result” factors. We did not find significant changes during the experiment. However, for energy in the delta range, we found a main effect of shooting results (*p* = 0.042) and cortex areas (*p* = 0.013) (see Figure 3D).

For energy in the alpha range, we did not reveal the main effects. Nevertheless, we found an interaction effect between cortex areas and shooting results (*p* = 0.016). In the post hoc analysis, we found significant changes in the right temporal lobe in the alpha range (*p* = 0.049507); however, a *p*-value was not adjusted for multiple comparisons.

Finally, we revealed that energies in the delta range and the alpha range in the right temporal lobe were significantly less before a hit compared to a miss.

### 3.3. Correlation Analysis

To identify the relationships between the characteristics under study, we performed a correlation analysis. The results of correlation analysis are shown in Table 1.

We discovered that changes in subjective fatigue positively correlate with average heart rate in the rest phase (r=0.42). Simultaneously, the hit rate correlates with the following parameters: respiration rate in the resting phase (r=0.33), respiration rate in the shooting phase (r=−0.35), and energies before the shot in the theta range in the frontal and central regions (r=−0.33 and r=−0.33, respectively).

Additionally, we identified the correlation between the NASA-TLX and the hit rate (ρ=−0.532).

## 4. Discussion

We analyzed multimodal psychophysiological data (EEG, ECG, EOG, respiration activity, and fatigue) to explore the neural and behavioral mechanisms underlying precision visual-motor control learning during sports shooting tasks. We systematically studied the relationship between physiological parameters, brain activity, and shooting performance over the course of learning to identify biomarkers that can be used to infer complex motor behavior. As expected, naive subjects significantly increased their hit rate during practice. Participants became, on average, ≈30% more accurate at shooting targets. Analysis of the physiological activity showed that performance improvements during the course of learning were accompanied by an increase in subjective fatigue and heart rate, wherein the average breathing rate remained unchanged. Respiration rate during shooting negatively correlates with marksmanship performance. We also found that the instantaneous respiration rate before a hit is higher than before a miss. Note that the work [17] did not reveal the influence of the instantaneous respiration rate on the shooting results. However, in study [18], the authors showed that respiration rate is related to the mental load, with high and medium load characterized by a significantly higher rate. In this regard, we hypothesize that our results may reflect a connection between mental load, instantaneous respiration rate, and shooting results. We suggest that in the case of a hit, the subjects were more deeply immersed and concentrated on the task and, accordingly, experienced a higher mental load than in the case of a miss.

Analysis of brain activity reveals several markers associated with shooting success. We found that average energy values in the delta range and the alpha range in the right temporal lobe were significantly less before a successful shot than before a miss. We identified that the hit rate negatively correlates with energies in the theta range in the frontal and central regions during the aiming period before shot execution.

The sport marksmanship task used in this study is one of the most convenient examples of tasks that can be used for investigating neurophysiological mechanisms underlying precise visual–motor coordination in a complex naturalistic context. Usually, studies addressing visual–motor integration by analyzing noninvasive recordings of cortical activity, such as EEG, involve laboratory tasks with minimal mobility to reduce artifact-producing muscle activity. Real-world tasks in natural environments require unrestricted full-body movements arising from full engagement of perception, decision-making, error recognition, and motor control. The shooting task is a controlled, easily replicated natural exercise that is particularly useful for investigating psychophysiological markers of visual-motor skill learning because it produces discrete measures of performance, which can be compared with electrophysiological activity recorded in real-time.

One of the main goals of this study was to identify EEG biomarkers of visual-motor skill learning during sport shooting tasks. Biomarkers are often referred to as quantitative indicators of a biological organism’s state and can be used to describe behavior-related psychophysiological processes. In recent years, the relationship of biomarkers with certain skills has been actively investigated [19,20,21]. The identified associations of skills with biomarkers are a promising tool for training process optimization.

In this study, we analyzed the relationship between EEG power in different frequency bands during the aiming period and shooting performance. We found that novices demonstrated delta and right temporal alpha EEG power increase before missing shots. Our results are in line with other studies reporting an overall reduction of alpha activity for experienced shooters [6]. This effect is interpreted as a greater engagement of task-relevant attentional processes. Janelle et al. [4] showed that shooting task expertise interacted with hemispheric activation levels. They demonstrated stronger alpha activity in the left hemisphere accompanied by its reduction in the right hemisphere for experts as compared to novices during the preparatory period before shot execution. Since shooting places high demands on visuospatial processing, the elevation of alpha power in the left temporal area may indicate a decrease of non-relevant to task cognitive activity (cognitive thinking, self-talk, or language analysis) and show that marksmen focused their attention on the visuospatial work dominated by right-brain areas [22,23,24].

Our results show the existence of a negative correlation between theta-band energies in the frontal and central brain regions during the preparation period and shooting success. Frontal midline (Fm) theta activation has often been observed in tasks that required consistent attention to a stimulus [25]. Recent studies reported Fm theta power as an indicator of sustained [26,27] and internalized [28,29] attention found in the preparation period in motor performance. Fm theta activity is linked to various kinds of attentional or working memory processes, such as working memory [30,31,32], learning [33], concentration [34], and action monitoring [35]. Sauseng et al. [26] associated Fm theta power with the number of cognitive resources allocated to attentional processes during a complex finger movement task learning. They clearly showed that Fm theta increased with increasing mental efforts and task demands. The results of our study are in line with these findings, demonstrating weaker theta activation with increasing correct acquisitions of the task and experience by the novice. Sport shooting task highly demands focused attention and precision visual-motor control. Shooting learning requires from the naive subjects a lot of cognitive resources and mental engagement. Therefore, the shooting training process is accompanied by a high level of mental effort reflected by increasing theta energy in the frontal and central brain regions. This explanation is confirmed by comparing the perceived workload level evaluated by NASA-TLX with the hit rate (see Table 1). Subjects with high hit rates reported greater confidence by feeling less workload level (lower levels of stress and pressure). In line with these results, Borghini et al. [36] demonstrated that the variation of the EEG power spectra in frontal areas in the theta band could be used as a measure for the training improvements of novices in flight simulation tasks. Their results showed that behavioral and task performance improvement was accompanied by a significant decrease in the theta band power over the frontal areas. Interestingly, the comparison of the time course of Fm theta during the aiming period in rifle shooting between experts and novices reveals that the theta power increased during the aiming process before the shot only for experts but not for novices [5]. The authors assume that elite marksmen are better able to allocate cortical resources in time while novices are unable to focus attention exactly on the shooting time point.

Note that this study has several limitations. First is the small number of participants (21). The second limitation is that only males participated in this study. Another limitation is using only EEG for the brain activity analysis since EEG has low spatial resolution compared to other techniques such as fMRI (functional magnetic resonance imaging). The last limitation is especially significant in the case of a possible investigation of visual-motor connection. For instance, in a recent paper [37], the usage of fMRI allowed researchers to discover a disrupted visual-motor connection in psychiatric disorders. In this study, however, fMRI is very difficult to use without substantial changes in the experimental paradigm.

## 5. Conclusions

In conclusion, our study sheds light on the neural and behavioral mechanisms underlying precision visual-motor control learning during sport shooting. We found that performance improvements were accompanied by an increase in subjective fatigue and heart rate and that the respiration rate before a hit was higher than before a miss, potentially reflecting a connection between the mental load and shooting results. Additionally, we identified several EEG biomarkers of visual-motor skill learning, including head-averaged delta and right temporal alpha EEG power increase before missing shots and a negative correlation between theta-band energies in the frontal and central brain regions and shooting success. The results of this study highlight the importance of considering both neural and behavioral factors in precision visual-motor control learning and the potential for using psychophysiological parameters to improve shooting performance. These findings provide valuable insights into the neurophysiological mechanisms underlying visual-motor skill learning and have potential implications for the optimization of training processes. 

## Figures and Tables

**Figure 1 sensors-23-03160-f001:**
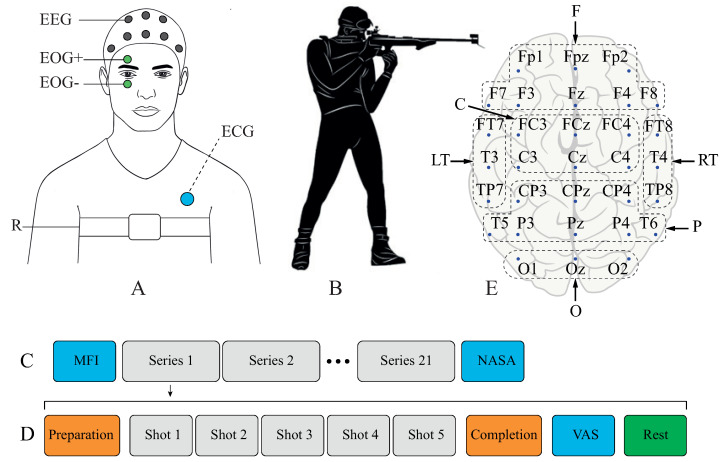
(**A**) Experimental setup with sensors: EEG (grey circles), EOG (green circles), ECG (blue circle), respiration (white stripe); (**B**) Shooting stance; (**C**) General design of the experimental session; (**D**) Design of individual series; (**E**) Scheme of EEG electrodes placement “10-10”. Chosen areas of EEG signals averaging are shown with dotted frames: frontal (F), central (C), parietal (P), occipital (O), left temporal (LT), and right temporal (RT).

**Figure 2 sensors-23-03160-f002:**
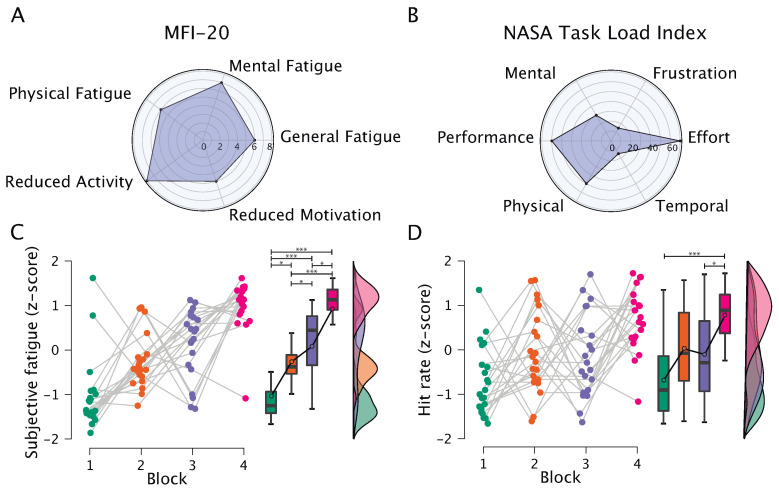
Results of the behavioral data analysis: (**A**) Median values for the scales of MFI-20 in the group of subjects; (**B**) Median values for the scales of NASA-TLX in the group of subjects; (**C**) Subjective fatigue (z-score); (**D**) Hit rate (z-score). Dots correspond to individual subjects, while box and whisker plots show values averaged over the blocks of the experiment. The symbol * denotes statistical significance in post hoc analysis using t-test with Holm’s correction for multiple comparisons (*—p<0.05, ***—p<0.001).

**Figure 3 sensors-23-03160-f003:**
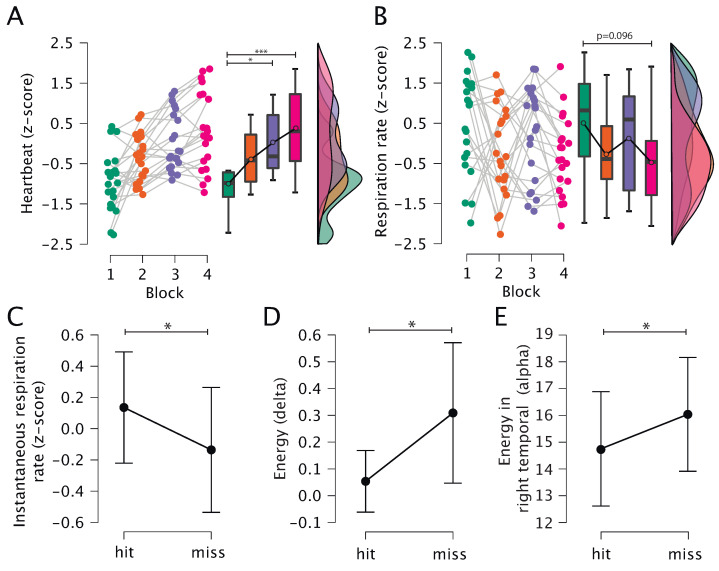
Results of the physiological data analysis: (**A**) Heartbeat at the rest phase (z-score); (**B**) Respiration rate during the shooting phase (z-score); (**C**) Instantaneous respiration rate corresponding to the shot time for hit and miss; (**D**) Average energy value in delta range before the shot for hits and misses; (**E**) Average energy value in the right temporal lobe in the alpha range before the shot for hits and misses. Dots correspond to individuals subjects while box and whisker plots show values averaged over the blocks of the experiment. The symbol * denotes statistical significance in post hoc analysis using t-test with Holm’s correction for multiple comparisons (*—p<0.05, ***—p<0.001).

**Table 1 sensors-23-03160-t001:** The results of correlation analysis.

	Hit Rate	Subjective Fatigue
Heart rate (resting phase)	-	r = 0.42, *p* = 0.006
Respiration rate (resting phase)	r = 0.33, *p* = 0.03	-
Respiration rate (shooting phase)	r = −0.35, *p* = 0.02	-
Energy (theta; frontal)	r = −0.33, *p* = 0.0073	-
Energy (theta; central)	r = −0.33, *p* = 0.0076	-
NASA-TLX	ρ = −0.532, *p* = 0.013	-

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
