# Peer review of "Psychophysiological Parameters Predict the Performance of Naive Subjects in Sport Shooting Training"

_sensors, 2023, doi:10.3390/s23063160_

Round 1
Reviewer 1 Report
The manuscript focused on an intersting topic and was generally written well. Below I have some concerns and suggestions to improve the manuscript.
There is only one figure in the section "2.2. Experimental Setup". It might be better to add some text to give more detailed explanations.
Why onle male participants included? Is there any reason?
It was mentioned that the study was approved by the local Research Ethics Committee. It may be better to give the exact name (and approval ID?) of the Ethics Committee.
During data acqusition and preprocessing, how to ensure data quality?
The age range is 18-30 in this sample. What is the average age? How did the authors control possible age effects in the analyzing models?
In correlation analyses, the authors should give p-values.
The authors didn't mention any limitaion of this study. In my opinion, the relatively small sample size (21 participants) should be considered as a limitation and can be mentioned.
Furthermore, the authors used EEG to detect brain activity in this study. However, the EEG is known to be limited by its relatively low spatial resolution compared with other techniques such as fMRI. Therefore it might be interesting to combine EEG and fMRI in the future studies, which can be mentioned in the limitation section. There have been actually some fMRI studies to investigate disrupted visual-motor connection in psychiatric disorders such as: doi.org/10.3389/fpsyt.2020.00422.
Author Response
Dear Editor,
First of all, we would like to thank the Referees and the Editor for their careful reading of our manuscript and for useful and very valuable comments, which we took into account in the revised version of the paper.
The Referees raised several comments, which we are addressing below. We also addressed the Editor’s comment about some parts of the manuscript with a high repetition rate.
The major changes in the revised manuscript are marked in blue. We hope that the current version of the manuscript is suitable for publication in the Sensors journal.
Best regards,
The Authors
Referee 1:
Comment
- There is only one figure in the section "2.2. Experimental Setup". It might be better to add some text to give more detailed explanations.
Answer
Thank you for the suggestion, we have added explanations to the section "2.2. Experimental Setup". We have added the description of the used recorder device, types of recorded signals, and placement of sensors with references to Figure 1.
Comment
- Why only male participants included? Is there any reason?
Answer
In our research, we excluded gender as a possible factor since gender difference is often observed in various works (for example, [Markovic, Andjela, Michael Kaess, and Leila Tarokh. "Gender differences in adolescent sleep neurophysiology: a high-density sleep EEG study." Scientific reports 10.1 (2020): 1-13.; Nishizawa, S., et al. "Differences between males and females in rates of serotonin synthesis in human brain." Proceedings of the national academy of sciences 94.10 (1997): 5308-5313]).
It is the limitation of this work, and in future research, we are planning to study possible gender differences as well.
Comment
- It was mentioned that the study was approved by the local Research Ethics Committee. It may be better to give the exact name (and approval ID?) of the Ethics Committee.
Answer
Thank you for the suggestion – we have added the exact name of the Ethics Committee and approval ID:
«This study was conducted according to the guidelines of the Declaration of Helsinki and approved by the Ethics Committee of Lobachevsky University (Protocol №3 from 8.04.2021).»
Comment
- During data acquisition and preprocessing, how to ensure data quality?
Answer
Thank you for your comment. We have expanded the "2.4. Data Processing" section to clarify the preprocessing procedure and its goals. We also stated the quality of data that we aimed at.
Comment
- The age range is 18-30 in this sample. What is the average age? How did the authors control possible age effects in the analyzing models?
Answer
Thank you for the question. It helped us to find a small inconsistency in our manuscript. The volunteers who participated in this study were between the age of 19-25, with an average age of 21 and a standard deviation of ~1.5. We believe that all participants belong to the same age group, so we do not expect any age effects.
Comment
- In correlation analyses, the authors should give p-values.
Answer
Thank you for the suggestion. We have added the p-values for all correlation results.
Comment
The authors didn't mention any limitaion of this study. In my opinion, the relatively small sample size (21 participants) should be considered as a limitation and can be mentioned. Furthermore, the authors used EEG to detect brain activity in this study. However, the EEG is known to be limited by its relatively low spatial resolution compared with other techniques such as fMRI. Therefore, it might be interesting to combine EEG and fMRI in the future studies, which can be mentioned in the limitation section. There have been actually some fMRI studies to investigate disrupted visual-motor connection in psychiatric disorders such as: doi.org/10.3389/fpsyt.2020.00422.
Answer
Thank you for the suggestion. We have added the discussion about the limitations of this study in the section "4. Discussion":
« Note that this study has several limitations. First, it is the small of number participants (21). The second limitation is that only males participated in this study. Another limitation is using only EEG for the brain activity analysis since EEG has low spatial resolution compared to other techniques such as fMRI (functional magnetic resonance imaging). The last limitation is especially significant in the case of a possible investigation of visual-motor connection. For instance, in a recent paper [37], usage of fMRI allowed to discover a disrupted visual-motor connection in psychiatric disorders. In this study, however, fMRI is very difficult to use without substantial changes in the experimental paradigm.»
[37] Long, Y.; Liu, Z.; Chan, C.K.Y.; Wu, G.; Xue, Z.; Pan, Y.; Chen, X.; Huang, X.; Li, D.; Pu, W. Altered temporal variability of local and large-scale resting-state brain functional connectivity patterns in schizophrenia and bipolar disorder. Frontiers in psychiatry 2020, 11, 422.
Reviewer 2 Report
The article is devoted to the analysis of the physiological parameters of novice shooters. From the wider spectrum of analyzed parameters, the greatest accent is placed on the EEG signal and determining the degree of fatigue. The aim of the authors was mainly to investigate the neural and behavioural mechanisms associated with precision visual-motor control during the learning of sport shooting.
The article meets all the necessary requirements, is comprehensible, clear, well-structured, and formatted. English, as far as I can judge, is at an appropriate level.
The selected experiments are appropriate. Data processing and statistics are done correctly, although I am not an expert in this area.
However, I have several comments, which I will summarize point by point:
The most serious shortcoming is the missing data in the methodology.
It will be necessary to add a precise description of hardware equipment and how they were placed. It is important to be able to assess whether, for example, the devices did not affect motor skills, did not cause discomfort, or were not a source of artifacts.
The shooting discipline and shooting conditions must be specified in detail. For example, the type of calibre used can significantly affect physiological data through noise/recoil. Protective gear such as shooting glasses/headphones can also play a role. Distance, type of sights (scope/dioptre/open sights) etc. are also important. Whether shooters had feedback after each shot, knew where they hit, or received feedback only after a series of shots. All these parameters can have an impact on the physiological data, and it is good that other researchers using this article in the future have this information.
I also partially miss the details of HR and RR (respiration rate) methodology. What long time windows were considered in the analysis? Was there an increased focus on shorter windows before the shot?
Was the EOG used in a more meaningful way, or just to determine the blinking rate?
In the list of authors are listed 3 workplaces but no one is associated with workplace 2.
The article itself as a good pilot study, and after incorporating the comments, I recommend accepting it, but I assume that the statistical relevance of the results is a little weaker. The reasons and recommendations to the author for the future I write again point by point:
The problem is that the chosen experiment is susceptible to too many input factors and artifacts that were not monitored. The fact that the authors chose novices as the shooting group is a wider spectrum of reasons for inaccurate hits. More experienced shooters can subjectively better assess the reason for their mistake, which can be included in the results and the data then divided into individual subgroups. Each reason for the error will have a different physiological response and each will have to be analyzed separately. I would choose the opposite procedure, start with experienced shooters, and gradually work my way up to beginners. I would supplement the experiment with the supervision of an experienced instructor, or at least partially analyzed the reason for the inaccuracy, the hitting zones. From the experience of shooting a pistol, I know that, for example, hits shifted on the target in the direction of 7-8 o'clock mean too fast pressing of the trigger, 6 o'clock anticipation and fear of recoil, etc. Something similar will happen when shooting a rifle, and the instructor will be able to identify it.
Rather, if the authors optimize the methodology and data analysis, they could even make shooting easier, e.g. to eliminate the motoric error of the shooting stance by shooting from the prone position, or even try shooting with a support, so that we can concentrate and identify inaccuracies due to poor tracking of the target, etc. Electronic shooting tracking systems such as www.scatt.com, www.traceshooting.com could be a great benefit.
In the future, I would recommend authors to focus mainly on a shorter period of time before the shot. It is also necessary to identify the moment of the shooting itself (microphone, accelerometer, etc.). HR or RR alone cannot have such a big impact, but rather HRV (heart rate variability) and respiration volume. I would focus more on these parameters than just pure long-term HR and RR.
Regarding the EEG processing itself, I would focus in more detail on the mapping of the sensory and motor cortex, specifically in SMR frequencies. I would also expect interesting results in the frontal lobe, which is associated with planning and concentration, specifically with beta waves or calculating the ratio of alpha/beta waves. Finally, I would also look at the occipital lobe of the dominant eye.
I believe that my comments will help authors to improve the article and improve the configuration of the experiment. I find the article very useful.
Author Response
Dear Editor,
First of all, we would like to thank the Referees and the Editor for their careful reading of our manuscript and for useful and very valuable comments, which we took into account in the revised version of the paper.
The Referees raised several comments, which we are addressing below. We also addressed the Editor’s comment about some parts of the manuscript with a high repetition rate.
The major changes in the revised manuscript are marked in blue. We hope that the current version of the manuscript is suitable for publication in the Sensors journal.
Best regards,
The Authors
Referee 2:
Comment
- It will be necessary to add a precise description of hardware equipment and how they were placed. It is important to be able to assess whether, for example, the devices did not affect motor skills, did not cause discomfort, or were not a source of artifacts.
Answer
Thank you for the suggestion. We have added the corresponding explanations to the section "2.2. Experimental Setup".
Comment
- The shooting discipline and shooting conditions must be specified in detail. For example, the type of calibre used can significantly affect physiological data through noise/recoil. Protective gear such as shooting glasses/headphones can also play a role. Distance, type of sights (scope/dioptre/open sights) etc. are also important. Whether shooters had feedback after each shot, knew where they hit, or received feedback only after a series of shots. All these parameters can have an impact on the physiological data, and it is good that other researchers using this article in the future have this information.
Answer
Thank you for very much this comment. We have added information about the shooting gear and conditions to “2.3 Experimental Procedure” section.
Comment
- I also partially miss the details of HR and RR (respiration rate) methodology. What long time windows were considered in the analysis? Was there an increased focus on shorter windows before the shot?
Answer
We have considered different time window scales for the analysis of HR and RR.
To find a difference between stages of the experiment (rest vs shooting), we averaged HR and RR in windows length equal to respective stages. The time length of windows for the resting stage is 60 seconds, but windows for the shooting stage have different lengths (average length of 22.5 seconds) because of different rates of shooting across the subjects and shooting stages. Also, we analyzed the influence of instantaneous (right at the moment of shot) RR on shooting results. For this, we calculated the instantaneous frequency of the signal from a series of peaks. It is calculated as “60/period”, where the period is the time between peaks. To interpolate the frequency over the entire duration of the signal, the monotone cubic interpolation method was used.
We have added the corresponding description to the section "2.4. Data Processing"
Comment
- Was the EOG used in a more meaningful way, or just to determine the blinking rate?
Answer
At this stage of the research, we considered just the blinking rate. In future work, we plan to conduct a more detailed analysis of EOG.
Comment
- In the list of authors are listed 3 workplaces but no one is associated with workplace 2.
Answer
Thank you for your comment, we have fixed this issue.
Comment
- The problem is that the chosen experiment is susceptible to too many input factors and artifacts that were not monitored. The fact that the authors chose novices as the shooting group is a wider spectrum of reasons for inaccurate hits. More experienced shooters can subjectively better assess the reason for their mistake, which can be included in the results and the data then divided into individual subgroups. Each reason for the error will have a different physiological response and each will have to be analyzed separately. I would choose the opposite procedure, start with experienced shooters, and gradually work my way up to beginners. I would supplement the experiment with the supervision of an experienced instructor, or at least partially analyzed the reason for the inaccuracy, the hitting zones. From the experience of shooting a pistol, I know that, for example, hits shifted on the target in the direction of 7-8 o'clock mean too fast pressing of the trigger, 6 o'clock anticipation and fear of recoil, etc. Something similar will happen when shooting a rifle, and the instructor will be able to identify it.
Rather, if the authors optimize the methodology and data analysis, they could even make shooting easier, e.g. to eliminate the motoric error of the shooting stance by shooting from the prone position, or even try shooting with a support, so that we can concentrate and identify inaccuracies due to poor tracking of the target, etc. Electronic shooting tracking systems such as www.scatt.com, www.traceshooting.com could be a great benefit.
In the future, I would recommend authors to focus mainly on a shorter period of time before the shot. It is also necessary to identify the moment of the shooting itself (microphone, accelerometer, etc.). HR or RR alone cannot have such a big impact, but rather HRV (heart rate variability) and respiration volume. I would focus more on these parameters than just pure long-term HR and RR.
Regarding the EEG processing itself, I would focus in more detail on the mapping of the sensory and motor cortex, specifically in SMR frequencies. I would also expect interesting results in the frontal lobe, which is associated with planning and concentration, specifically with beta waves or calculating the ratio of alpha/beta waves. Finally, I would also look at the occipital lobe of the dominant eye.
Answer
Thank you so much for your interest in our work and important comments.
We agree that our experiment includes factors that were not monitored and for this reason, we may not have detected subtle effects associated with misses. However, in this study, we focused on the investigation of more pronounced effects associated with intensive training over time for novice shooters. We are planning to analyze various causes of misses in detail in future research, and electronic shooting tracking systems such as www.scatt.com and www.traceshooting.com could be used for a great benefit.
Thank you again for your valuable comments. We will take them into account in our next research.
Round 2
Reviewer 2 Report
Good job.
Thanks for incorporating my comments. I recommend the article for publication and wish you success in your future work.
If I ever in the future start to devote myself more intensively to a similar issue, I would like to establish cooperation with you.